# Exploring the Effect of Acute and Regular Physical Exercise on Circulating Brain-Derived Neurotrophic Factor Levels in Individuals with Obesity: A Comprehensive Systematic Review and Meta-Analysis

**DOI:** 10.3390/biology13050323

**Published:** 2024-05-06

**Authors:** Halil İbrahim Ceylan, Ana Filipa Silva, Rodrigo Ramirez-Campillo, Eugenia Murawska-Ciałowicz

**Affiliations:** 1Physical Education and Sports Teaching Department, Kazim Karabekir Faculty of Education, Ataturk University, 25240 Erzurum, Turkey; 2Escola Superior Desporto e Lazer, Instituto Politécnico de Viana do Castelo, Rua Escola Industrial e Comercial de Nun’Álvares, 4900-347 Viana do Castelo, Portugal; 3Research Center in Sports Performance, Recreation, Innovation and Technology (SPRINT), 4960-320 Melgaço, Portugal; 4Exercise and Rehabilitation Sciences Institute, School of Physical Therapy, Faculty of Rehabilitation Sciences, Universidad Andres Bello, Santiago de Chile 7591538, Chile; rodrigo.ramirez@unab.cl; 5Department of Physiology and Biochemistry, Faculty of Physical Education and Sport, Wrocław University of Health and Sport Sciences, 51-612 Wrocław, Poland; eugenia.murawska-cialowicz@awf.wroc.pl

**Keywords:** obesity, BDNF, physical exercise, meta-analysis

## Abstract

**Simple Summary:**

Obesity is associated with cognitive impairment and reduced levels of circulating brain-derived neurotrophic factor (BDNF), a protein crucial for brain function and health. The aim of this systematic review was to overview the effects of acute (a single session) and regular (long-term) exercise on circulating BDNF levels in obese individuals. The meta-analysis of 16 studies with 23 trials revealed an increase in BDNF levels after a single session of exercise in individuals with obesity. However, long-term exercise did not elevate circulating BDNF levels. These findings highlight the complexity of the relationship between exercise, obesity, and brain health. Further research is needed to delve deeper into how different exercise parameters, such as type, duration, and intensity, impact BDNF levels in obese individuals. Understanding these nuances can help tailor exercise interventions more effectively to improve brain function and overall well-being in this population.

**Abstract:**

Obesity is a major global health concern linked to cognitive impairment and neurological disorders. Circulating brain-derived neurotrophic factor (BDNF), a protein crucial for neuronal growth and survival, plays a vital role in brain function and plasticity. Notably, obese individuals tend to exhibit lower BDNF levels, potentially contributing to cognitive decline. Physical exercise offers health benefits, including improved circulating BDNF levels and cognitive function, but the specific impacts of acute versus regular exercise on circulating BDNF levels in obesity are unclear. Understanding this can guide interventions to enhance brain health and counter potential cognitive decline in obese individuals. Therefore, this study aimed to explore the impact of acute and regular physical exercise on circulating BDNF in individuals with obesity. The target population comprised individuals classified as overweight or obese, encompassing both acute and chronic protocols involving all training methods. A comprehensive search was conducted across computerized databases, including PubMed, Academic Search Complete, and Web of Science, in August 2022, following the Preferred Reporting Items for Systematic Reviews and Meta-Analyses (PRISMA) guidelines. Initially, 98 studies were identified, from which 16 studies, comprising 23 trials, met the selection criteria. Substantial heterogeneity was observed for both acute (I^2^ = 80.4%) and long-term effects (I^2^ = 88.7%), but low risk of bias for the included studies. A single session of exercise increased circulating BDNF levels among obese patients compared to the control group (ES = 1.25, 95% CI = 0.19 to 2.30, *p* = 0.021). However, with extended periods of physical exercise, there was no significant increase in circulating BDNF levels when compared to the control group (ES = 0.49, 95% CI = −0.08 to 1.06, *p* = 0.089). These findings highlight the need to consider exercise duration and type when studying neurobiological responses in obesity and exercise research. The study’s results have implications for exercise prescription in obesity management and highlight the need for tailored interventions to optimize neurotrophic responses. Future research should focus on elucidating the adaptive mechanisms and exploring novel strategies to enhance BDNF modulation through exercise in this population. However, further research is needed considering limitations such as the potential age-related confounding effects due to diverse participant ages, lack of sex-specific analyses, and insufficient exploration of how specific exercise parameters (e.g., duration, intensity, type) impact circulating BDNF.

## 1. Introduction

The World Health Organization (WHO) stated that, worldwide, obesity has nearly tripled since 1975 [1]. Despite being deemed preventable, the ramifications of being overweight and obesity are pervasive, impacting almost 60% of adults and nearly one in three children, with 29% of boys and 27% of girls affected, within the WHO European Region. Currently, obesity is considered a disease by several medical institutions (e.g., The American Medical Association; WHO; Obesity Society), and the World Obesity Federation sees obesity from an epidemiological model, i.e., as an agent affecting the host and producing disease [2]. In fact, it is now clear from long-term follow-up studies that obesity is related to a greater probability of developing heart diseases, type 2 diabetes mellitus, some cancers, dementia, osteoarthritis, among others [3,4,5,6]. Recently, attention has been given to the link between obesity and neurobiological impairments, since the high levels of dysfunctional adipose tissue observed in patients with obesity may aggravate metabolic abnormalities, which increase the risk of mood disorders as depression [7,8] and affect the balance of energy expenditure control [9,10]. Indeed, the brain-derived neurotrophic factor (BDNF), which is a protein that has a significant role in the energy homeostasis of body fluids and blood pressure in humans [11], registered significantly lower levels in the patients with obesity [7,12]. Reduced satiety and hyperphagia manifest as discerning features of BDNF deficiency, providing a plausible rationale for fat accumulation. This phenomenon is undeniably associated with the pivotal role of BDNF in the intricate regulation of dietary intake [13]. Furthermore, diminished levels of circulating BDNF have been ascertained in individuals diagnosed with type 2 diabetes [14]. Notably, an inverse correlation has been established between peripheral BDNF concentration and key anthropometric parameters, such as body mass index (BMI), in both pediatric and adult populations [15]. Additionally, in adult males, a negative relationship has been identified between peripheral BDNF levels and fat mass [16].

Indeed, circulating BDNF supports survival, growth, and maintenance of neurons during development [17,18], and influences synaptic plasticity in the adult brain [19]. The impact of circulating BDNF encompasses developmental processes, the regulation of neuron and glial cell formation, the protection of neurons, and the modulation of synaptic connections that affect memory and cognition mechanisms [20,21,22]. In opposition, deletion or inhibition of the BDNF gene results in a deficiency in long-term potentiation, a transcription-dependent electrophysiological phenomenon associated with learning and memory [20,22,23]. This synaptic dysfunction may be corrected by external administration of BDNF [18] or by increasing its expression [24].

Physical exercise may improve circulating BDNF concentration, thus inducing brain plasticity and cognitive enhancement [25,26]. Physical exercise might favor the release of neurotransmitters and neurotrophins in an activity-dependent manner. This acute stimulation potentiates neural function and initiates a cascade of events that actively contribute to the promotion of structural and functional plasticity within the brain [27,28,29]. Physical exercise elevates the rate of mitochondrial respiration and cerebral oxygen consumption [30], with heightened levels of circulating BDNF [31,32,33] and enhanced functionality of the prefrontal cortex in individuals without pre-existing health conditions [34].

The role of circulating BDNF in neurological impairments associated with obesity has received considerable attention in the recent literature. Indeed, circulating BDNF has been implicated in the pathophysiology of several neurological disorders, including cognitive decline associated with obesity [35,36]. Decreased BDNF levels correlate with cognitive deficits often seen in obese individuals, including impaired learning and memory, executive dysfunction, and mood disorders [37,38]. Likewise, interventions that boost BDNF expression, such as physical exercise, have been associated with enhanced cognitive performance and mood in both healthy individuals and those grappling with obesity [36]. Moreover, previous studies conducted on obese animals have offered mechanistic insights into how circulating BDNF regulates neuronal function and synaptic plasticity in brain regions essential for cognitive processes, such as the hippocampus and prefrontal cortex [39,40]. These findings highlight circulating BDNF’s significance as a potential therapeutic target for alleviating neurological complications associated with obesity. Despite the expanding body of evidence, the precise mechanisms by which circulating BDNF exerts its neuroprotective effects in the context of obesity remain partially understood. Nonetheless, factors such as obesity-related inflammation, insulin resistance, and alterations in neurotrophin signaling pathways may contribute to circulating BDNF dysregulation and subsequent cognitive impairments [35]. The link between obesity and altered BDNF signaling underscores the importance of understanding how physical exercise, a primary intervention for obesity management, affects circulating BDNF levels [41]. Furthermore, while emerging evidence suggests that physical exercise can modulate circulating BDNF levels, its impact on individuals with obesity remains to be fully elucidated. Therefore, a comprehensive understanding of how both acute and regular physical exercise influence circulating BDNF levels in this population is warranted.

Given the intricate interplay between obesity, physical exercise, circulating BDNF, and neurological function, this study aims to systematically review and meta-analyze studies related to the effects of acute and regular physical exercise on circulating BDNF levels in individuals with obesity. By synthesizing the existing literature and elucidating the relationship between exercise, circulating BDNF, and neurological outcomes, we aim to offer insights that can guide future interventions aimed at preserving cognitive health in this population. Our hypothesis suggests that physical exercise has the potential to elevate circulating BDNF levels in individuals with obesity, thus potentially alleviating neurobiological impairments associated with the condition and enhancing brain health.

## 2. Methods

The present review adhered to the Preferred Reporting Items for Systematic Reviews and Meta-Analyses (PRISMA) 2020 guidelines [42,43].

### 2.1. Protocol and Registration

The systematic review protocol is accessible on the Open Science Framework (OSF) under the registration number DOI 10.17605/OSF.IO/4UM3B, documented on the 10th of August 2022. Interested parties can review the protocol through the following web address: https://archive.org/details/osf-registrations-4um3b-v1 (accessed on 30 April 2024).

### 2.2. Eligibility Criteri

Research articles considered for inclusion in this study were published in peer-reviewed journals, without imposing any restrictions on the publication date. The eligibility criteria are constructed according to the Participants, Intervention, Comparators, Outcomes, and Study Design (PICOS) framework below [44,45]:

Participants (P): As a criterion for inclusion, studies involving subjects classified as overweight (Body Mass Index-BMI = 25–30 kg/m^2^) or obese (BMI = 30 kg/m^2^ or greater) according to the BMI have been accepted. This instrument was chosen because most studies reported BMI values rather than using more specific measurements for assessing body composition. The classification employed in the original studies were disregarded. Studies conducted with participants of both genders, spanning any age within range, have been included. Studies involving athletes or those incorporating subjects with normal body weight have been excluded.

Intervention (I): Both acute (single-session) and regular physical exercise interventions (multiple sessions, with no minimum session requirement) including aerobic, high intensity interval training, resistance training, or combined training methodologies were included. Studies lacking physical exercise programs or containing only cognitive programs were excluded.

Comparators (C): In acute effect studies, the comparator encompassed pre-and post-evaluations. For interventions, comparators involved at least one experimental group or the inclusion of a control group.

Outcomes (O): The primary outcome of interest is circulating BDNF variability, with a focus on the methodology applied during the session or sessions. Furthermore, we included studies that specifically assessed circulating levels of BDNF in serum or plasma samples.

Study Design (S): Only experimental studies, whether acute or interventions, and whether randomized or non-randomized clinical trials, were considered for inclusion [44,45].

### 2.3. Data Sources and Search Strategy

The current systematic review involved searching the following databases on the same day (05/08/2022): (i) PubMed; (ii) Academic Search Complete; and (iii) Web of Science. The search encompassed files up to the present year, with no lower limit. Additionally, a manual search was carried out to identify potentially relevant articles not covered in the automated searches, including sources such as Google Scholar and ResearchGate. The manual search focused on: (i) scrutinizing the reference lists of included full texts to identify potentially relevant titles; (ii) reviewing abstracts for adherence to inclusion criteria; and, if necessary, (iii) revising the full text. Furthermore, errata/retractions were examined to ensure the accuracy of the included articles [46]. The search strategy employed Boolean operators AND/OR without applying any filters or limitations (e.g., date, study design) to enhance the likelihood of identifying relevant studies [47]. The primary search strategy comprised the following terms:

“BDNF” OR “brain-derived neurotrophic factor”

AND

“aerobic*” OR “HIIT” OR “high intensity interval training” OR “anaerobic*”

AND

“obesity” OR “overweight”

The complete search strategy [47] is detailed in Table 1

### 2.4. Selection Process

The initial phase of the study involved a screening process conducted by two authors, namely HC and AFS, who independently evaluated the titles and abstracts of the retrieved records. Following this initial screening, the same authors independently assessed the full texts of the gathered literature. Discrepancies that arose between these two authors were subjected to thorough discussion in a collaborative reanalysis. In instances where a consensus could not be achieved, a third author, EMC, resolved.

### 2.5. Data Collection Process

The data collection process commenced with AFS as the primary investigator (November 2022), who meticulously gathered the data. To ensure the accuracy and completeness of the collected information, a dual verification process was implemented, involving two co-authors, namely HC and EMC. A specially designed Microsoft^®^ Excel datasheet was employed as the tool for extracting and organizing the data, encompassing key information pertinent to the study [48]. In instances where essential data were absent from the full text of the included studies, proactive measures were taken by author HC, who initiated direct communication with the corresponding authors via email and/or ResearchGate. This communication sought to obtain the requisite information, with an anticipated response time of approximately 10 business days. The systematic extraction of pre- and post-intervention means, coupled with the standard deviation of dependent variables was methodically carried out using Microsoft Excel (Microsoft Corporation, Redmond, WA, USA) [48]. In situations where studies reported data in formats other than means and standard deviations—specifically, presenting values such as median, range, interquartile range, or standard error—a methodically standardized conversion procedure was systematically implemented, strictly adhering to established recommendations [49,50,51]. The statistical analysis of diverse data formats employed the Comprehensive Meta-Analysis Software, Version 2, developed by Biostat in Englewood, NJ, USA. This software was chosen for its versatility in handling various data structures inherent in the included studies [52]. In instances where the required data were not exhaustively reported in the literature, the authors of the respective studies were engaged through direct communication to seek clarification. This proactive approach aimed to enhance the accuracy and completeness of the dataset under consideration. Following a standard protocol, if no response was received from the authors after two attempts, with 10 business days waiting period between attempts, or if the authors were unable to provide the requested data, the study outcome was deemed ineligible for further analysis. Moreover, when data were visually presented in figures without accompanying numerical values, a validated software tool, WebPlotDigitizer, version 4.5 (Rohatgi, A., Pacifica, CA, USA. https://apps.automeris.io/wpd/, accessed on 30 April 2024), was employed. This software, validated with a correlation coefficient (r = 0.99, *p* < 0.001) [53], facilitated the extraction of numerical data from the graphical representations. The extraction process was spearheaded by one author, AFS, with another author, HC, responsible for confirming the accuracy of the extracted data. Any discrepancies between the two authors, such as disagreements on mean values for specific outcomes, were judiciously resolved through consensus with a third author, ensuring the reliability and consistency of the data extraction process throughout the analytical phase [54].

### 2.6. Data Items

Descriptive characteristics of study participants, including sex and physical activity level were documented to provide a contextual understanding of the study population.

Context-related information played a pivotal role in the data extraction process and included, though was not restricted to, the presence of other clinical complications such as diabetes and hypertension, among others.

Methodological-related details were integral to the analysis and included information on the specific physical exercise protocols employed in the studies. This encompassed details such as the type of exercise (aerobic, HIIT, resistance, combined training), as well as the number, volume, and intensity of the exercise sessions.

The main outcome of interest, namely the changes in circulating BDNF in response to the prescribed physical exercise protocols, formed the core focus of the data extraction process. Variations in circulating BDNF levels, whether indicative of increases or decreases, were systematically recorded to discern the effects of different exercise methodologies.

Additionally, supplementary information was gathered, including citation details, the publication year, and any potential competing interests declared by the authors.

### 2.7. Risk of Bias Assessment

The evaluation of the risk of bias within each study was conducted independently by two authors, namely EM and AFS. In instances where discrepancies emerged between the two assessors, a collaborative reanalysis was undertaken to resolve differences. In instances where a consensus could not be attained, a third author resolved. The Risk of Bias Assessment Tool for Non-randomized Studies (RoBANS) was employed as the standardized instrument for evaluating the risk of bias in the included studies [55]. This tool has demonstrated moderate reliability, as well as trustworthy feasibility and validity [55], making it a suitable choice for the present analysis. The RoBANS framework encompasses six pivotal domains: participant selection, confounding variables, exposure measurement, outcome assessment blinding, handling of incomplete outcome data, and avoidance of selective outcome reporting [55]. Each of these domains was systematically assessed, and the risk of bias within each domain was classified as low, high, or unclear.

### 2.8. Data Management and Synthesis Methods

The analytical approach adhered to a previously established methodology, as outlined in references [56,57], wherein the analysis and interpretation of results were undertaken only when a minimum of three studies provided both baseline and follow-up data for the same outcome measure. The effect size (ES), denoted by Hedge’s g, was computed for each outcome measure within the experimental groups by utilizing pre- and post-exercise mean values in conjunction with standard deviations (SD) [49,50]. Standardization of data involved the utilization of post-intervention SD values. A random-effects model was employed to accommodate variations between studies that might influence the impact of interventions on BDNF response [58,59]. Effect sizes were presented with 95% confidence intervals (CIs) and interpreted as follows: <0.2, trivial; 0.2–0.6, small; >0.6–1.2, moderate; >1.2–2.0, large; >2.0–4.0, very large; >4.0, extremely large [59]. The assessment of heterogeneity was conducted using the I^2^ statistic, with low heterogeneity characterized by values < 25%, moderate heterogeneity when values fell between 25% and 75%, and high heterogeneity observed with values > 75% [60]. Publication bias was evaluated utilizing the extended Egger’s test [58]. In instances where bias was identified, the trim and fill method was applied, with L0 serving as the default estimator for missing studies [61,62]. All statistical analyses were performed using Comprehensive Meta-Analysis software (version 2; Biostat, Englewood, NJ, USA) [52]. Statistical significance was set at *p* ≤ 0.05. Moderators associated with acute or interventional studies, physical exercise frequency, and type were considered in cases where two or more studies provided relevant data, enhancing the depth of the analysis and interpretation of results.

## 3. Results

### 3.1. Study Identification and Selection

The initial phase of the literature search yielded 98 titles, which were systematically managed using the EndNote^TM^ reference manager software (version 20.2, Clarivate Analytics, Philadelphia, PA, USA). After removal of duplicates, totaling 35 titles, through an automated and manual process, the dataset was refined to 63 unique titles. These were then subjected to a screening process based on their relevance, involving an assessment of the title and abstract. This screening led to the exclusion of 32 titles. The remaining 31 titles underwent a thorough evaluation in their full-text versions, resulting in the exclusion of an additional 15 studies for various reasons. Specifically, one study focused on pregnant individuals, five studies did not incorporate physical exercise and were merely report-based, one study constituted a review paper, and eight studies were excluded due to issues related to the outcome measure. In the latter case, one study evaluated proBDNF instead of mBDNF, while in the remaining eight studies, no pertinent data were available in the manuscripts, and attempts to obtain information from the authors were unsuccessful. Following this selection process, 16 studies, encompassing 23 trials remained eligible for data extraction and subsequent analysis. This final set of studies formed the basis for the comprehensive examination and synthesis of data to address the research objectives (Figure 1).

### 3.2. Study Characteristics and Context

The characteristics and contexts of studies showing the effect of acute and regular exercise on circulating BDNF are shown in Table 2, Table 3 and Table 4.

### 3.3. Risk of Bias in Studies

The Risk of Bias Assessment Tool for Non-Randomized Studies (RoBANS) was applied by two independent authors to evaluate potential biases in six distinct do-mains. Notably, the selection of participants posed challenges in several studies, as the recruitment methods were not consistently explained, raising concerns about potential convenience sampling without clear contextual information. However, given that most studies adhered to clinical trial assumptions, the domains of confounding variables, exposure measurements, and blinding of outcome assessments exhibited a low risk of bias. The comprehensiveness of protocols was generally high, with 16 studies providing detailed explanations, while a few studies lacked minor protocol characteristics. One study lacked information regarding training load characteristics. Regarding outcome data, biases were identified in some studies (three at high risk), where explanations for discrepancies in sample sizes from the beginning to the end of the study were not provided. In the overall analysis, the included studies appeared to have a low risk of bias, with none exhibiting a high risk in more than three domains. Despite uncertainties in participant selection methodologies in some studies, the majority adhered to rigorous clinical trial principles in other key domains, contributing to the overall robustness of the studies. These findings underscore the methodological quality of the included studies in this assessment of bias risk.

### 3.4. Results of Syntheses

#### Effect of Acute and Regular Exercise on Circulating BDNF Level

This systematic review with meta-analysis synthesized findings from three studies encompassing a total of six trials, involving 104 participants, to assess the impact of acute exercise on circulating BDNF levels (see Figure 2). The meta-analysis revealed a statistically significant increase in circulating BDNF levels in individuals with obesity compared to controls (Effect Size, ES: 1.25, large effect, *p* < 0.05). However, there were indications of heterogeneity in the results (I^2^ = 80.4%).

Thirteen studies, encompassing 17 trials and involving a total of 571 participants, investigate the impact of regular physical exercise interventions on circulating BDNF levels (see Figure 3). Regular exercise interventions did not change circulating BDNF levels when compared to control groups (*p* > 0.05). Despite the lack of statistical significance, a small positive effect on circulating BDNF levels was observed in the exercise intervention group (effect size, ES: 0.49, small effect). It was determined high heterogeneity among the studies investigating the impact of regular exercise intervention on circulating BDNF levels (I^2^ = 88.7%).

## 4. Discussion

The aim of this systematic review with meta-analysis was to understand the acute (short-term) and regular (long-term) effects of physical exercise on circulating BDNF levels in patients with obesity. Acute exercise led to a significant increase in the concentration of circulating BDNF patients with obesity compared to the control group. Despite long-term regular physical exercise, there was no significant increase in circulating BDNF levels when compared to the control group. Although the results were not statistically significant, it was revealed a small effect size of long-term physical exercise interventions on circulating BDNF levels. (ES = 0.49). Our study also found high levels of heterogeneity in both acute effects (I^2^ = 80.4%) and long-term effects (I^2^ = 88.7%). Finally, the quality of the methodologies used in the included studies was determined to be low risk of bias, with none of the studies exhibiting high risk in more than three of the six evaluated domains.

Acute physical exercise has been shown to be an effective stimulant that arise peripheral BDNF levels [79]. The present study found that acute exercise led to a statistically significant augment in circulating BDNF level compared to the control group (ES: 1.25). This review synthesized the findings of three studies involving 104 participants to specify the impact of acute physical exercise on BDNF levels. This result confirms previous systematic reviews and meta-analysis studies showing that acute exercise increases circulating BDNF levels in healthy adults [36,41,79,80], and older adults [81]. Regarding the included studies, Dominguez-Sanchéz et al. [63] found that acute combine exercise (high-intensity interval exercise and resistance) led to a greater increases (+11.6%, *p* = 0.029) in circulating BDNF levels compared with acute resistance (+9.3%, *p* = 0.066), and high intensity interval exercises (+6.8%, *p* = 0.134) in overweight men adults. Likewise, another study notified that acute high-intensity exercise (20 min at 85% of VO_2_max) triggered a rise in serum BDNF levels in inactive adult patients with obesity [64]. Additionally, Wheeler et al. [65] (2020) observed that 30 min of aerobic exercise (65% and 75% HRmax) performed in the morning hours elevated circulating BDNF levels in inactive elderly men patients with obesity and postmenopausal women. Based on the studies mentioned (3 studies, 6 trials), regardless of the type of exercise, it has been shown that circulating BDNF levels increase in patients with obesity after both acute moderate (1 trial) and high-intensity exercises (5 trials).

Considering the studies mentioned above (3 studies, 6 studies), regardless of the type of exercise (aerobic, resistance and high intensity exercise), it is seen that the circulating BDNF levels of patients with obesity increase after both acute moderate (1 trials), and high-intensity exercise (5 trials). Regarding the studies analyzed, high-intensity exercise protocols were commonly used in patients with obesity, resulting in elevated circulating BDNF levels. The elevation in circulating BDNF levels after acute exercise can be linked to exercise intensity. For instance, previous systematic reviews indicated that circulating BDNF levels augmented with increased intensity [36,82] and duration (lasting more than 30 min) of the exercise [80]. Furthermore, a recent meta-analysis of 22 studies in healthy adults (aged 20–31 years) with a total of 552 participants noted that greater circulating BDNF levels were observed after acute high-intensity exercise compared to light and moderate-intensity exercise conditions [83].

Several potential mechanisms behind the elevation of circulating BDNF after acute exercise have been proposed. One of these is exercise-induced thrombocytosis (EIT). Due to the fact that the most of peripheral BDNF is found in thrombocytes, exercise can lead to a rise in BDNF levels through EIT, which is thought to occur due to splenic contractions releasing BDNF-rich platelets [84,85]. However, the increase in platelets and peripheral blood BDNF after exercise was reported to be temporary and returned to pre-exercise levels within 15–30 min after cessation of the physical activity [86]. Despite its temporary nature, it was stated that the increased BDNF response to acute exercise could have the potential to enhance cognitive function by triggering various neuronal processes [87]. Second, the contraction of skeletal muscle during physical activity boosts the manufacture and activity of certain proteins involved in mitochondrial biogenesis, such as PGC-1α, ERRalpha, and fibronectin type III domain-containing protein (FNDC5)/irisin. These factors regulate BDNF transcription in the brain and energy metabolism in skeletal and fat tissue [88]. Third, exercise elevates insulin-like growth factor-1 (IGF-1). This hormone has been shown to have a close relationship with an increase in BDNF level in the hippocampus, playing a role in mediating exercise-induced changes in synaptic function and cognitive plasticity [89]. Lastly, a single high-intensity exercise has been linked to higher levels of brain hydrogen peroxide (H_2_O_2_) and tumor necrosis factor-α (TNF-α). These molecules are among the many stimulators of PGC-1α signaling, which in turn increases BDNF synthesis in neurons [90].

Furthermore, circulating BDNF, like leptin, promotes feelings of satiety. Recent studies has shown that it actively controls food intake, regulates body weight, and balances energy at the hypothalamic level [91,92]. According to these studies, the BDNF-producing neurons in the paraventricular hypothalamus were found to limit food intake and serve as an anorexigenic factor, resulting in feelings of satiety. These neurons also improve energy consumption by stimulation of thermogenesis in brown adipose tissue. Also, mutations in the BDNF gene and its receptor TrkB in mice and humans caused an increase in food consumption, and contributed to the onset of severe obesity [93,94]. Based on these studies, the rise of circulating BDNF levels, which are lower baseline in patients with obesity compared to normal-weight individuals [94,95,96], following high-intensity exercise may be related to decreased hunger and an increased sense of fullness. This can minimize the likelihood of obesity developing arising from excessive appetite sensation or appetite signal disorder, and in addition, it can make an additional contribution to the treatment of various neuropsychological diseases accompanying obesity. More mechanisms of neuropsychological disturbance, especially related to the depression state has been summarized by Murawska-Cialowicz et al. [97]. Moreover, a previous study noted that administering BDNF to mice increased the levels of GLUT4 in skeletal muscle [98]. This suggests that the possible impact of increased BDNF levels following acute high-intensity exercise on substrate utilization may enhance the management of metabolic disorders related to obesity [99]. Obesity is a metabolic disorder associated with inflammation and also poses a serious risk of developing cardiovascular diseases, including hypertension [100,101]. Endothelial dysfunction in these conditions leads to the development of atherosclerosis [102]. Reduced production of nitric oxide (NO) by reactive oxygen species (ROS), inflammation, imbalance between vasodilators and vasoconstrictors, and blood vessels are important mechanisms leading to endothelial dysfunction [103]. BDNF production by skeletal muscle during exercise is thought to originate specifically from endothelial cells and is stimulated by NO, especially in oxidative fibers, although greater BDNF expression is observed in type II glycolytic fibers [104].

Numerous studies indicate that there is cross-talk between organs and that substances in the nature of growth factors, as BDNF, which is one of the most important factors communicating skeletal muscle with brain and adipose tissue, are involved in this communication. Therefore, the term, metabolokine, is used in relation to BDNF, which is a neurokinin, adipokine and myokine [105]. This is the way one of the possible mechanisms of BDNF synthesis during high intensity exercise, is the production of lactate [La-]. High-intensity exercise stimulates anaerobic energy processes. The product of this process, especially glycolysis is high concentration of [La-] that exceeds the anaerobic threshold. Nowadays, [La-] is known to be a very important substance acting as a transmitter involved in various metabolic processes [106,107,108]. In the brain [La-] W mediated by monocarboxylate transporters (MCT) [109,110] enters into neurons (via MCT2) and astrocytes (via MCT4) supporting, among other things, glucose transport into neurons, their energy metabolism and ion [111].

Current studies indicate that [La-] transport from astrocytes to neurons plays a key role in memory formation [112,113] and may represent a link between exercise and neuroplasticity and BDNF synthesis [107]. In this mechanism, [La-] can activate of various G protein-related receptors, as well as the silent information regulator 1 (SIRT1). With relation to adipose tissue, activation of the PGC1α/FNDC5/BDNF pathway seems to be a convincing one [114]. Lactate can induce the PGC1α/FNDC5/BDNF pathway through activation of SIRT1. Intraperitoneal infusion of lactate in mice was shown to induce SIRT1 activity, thereby enhancing the PGC1α/FNDC5/BDNF pathway, resulting in improved spatial learning and memory retention [115]. It was documented that lactate in the adipocytes in obesity is a key player connecting obesity with inflammation and insulin resistance and that the higher adipocytes size is related to the high lactate production in adipose tissue [116] which is resulted in hypoxia in obese adipocytes [117]. Moreover lactate can stimulate browning of the white adipocytes [118]. In white adipose tissue (WAT), MCT-1 is expressed in white adipose adipocytes and may be treated as a marker of adipocytes maturation [119].

FNDC5 is a precursor protein called irisin, which is an exercise hormone involved in carbohydrate and fat utilization and fat reduction. Expression/secretion of irisin promotes the conversion of WAT into brown adipose tissue (BAT) (browning) due to increased expression of UCP-1 (uncoupling protein 1). WAT browning was found to be induced by irisin through p38 MAPKs and ERK MAPK signaling [120]. It was documented that lack of irisin is associated with poor adipocyte browning and impaired glucose/lipid levels [121]. The another study shows that irisin concentrations are increased after intense exercise in healthy men [122] as well as in obese subjects in whom aerobic exercise showed no change in irisin levels [123] which given the common mechanism of action of irisin with BDNF, may potentiate BDNF secretion during exercise.

Chronic physical activity increases circulating BDNF in answer to an acute exercise [41]. Regarding the relationship between regular exercise intervention and circulating BDNF, this systematic review and meta-analysis summarized the findings of 13 studies comprising a total of 571 participants, regarding the impact of regular exercise intervention on circulating BDNF levels. The associations between regular exercise interventions and circulating BDNF is complicated [85]. In the literature, recent meta-analysis studies indicated that peripheral BDNF concentrations significantly increased after exercise intervention in healthy individuals [36,124], older adults [81], individuals with multiple sclerosis [125], and neurodegenerative disorders [126]. Contrary to the studies mentioned, the present study observed that regular physical exercise did not significantly augment circulating BDNF levels compared to the control group. Although not statistically significant, it was found a small effect size of regular physical exercise on circulating BDNF levels (ES = 0.49; trivial, *p* = 0.089). Consistent with our study, previous meta-analysis studies demonstrated that chronic exercise had a minimal impact on circulating BDNF level in healthy [41,80] and elderly individuals [127]. The current meta-analysis of included studies showed inconsistent findings considering the effect of chronic physical activity on circulating BDNF level. According to this, it was notified that circulating BDNF level significantly increased [66,67,71,72,74,75,76,77], decreased [68,69], and remained unchanged [70,73,78] after exercise intervention in patients with obesity. However, the lack of significant changes in circulating BDNF levels following regular exercise in our meta-analysis could be due to the high heterogeneity in the studies included, which explains the inconsistent results. Moreover, the high heterogeneity could be attributed to differences in the methodology of the included studies, including the characteristics of participants, exercise applications (frequency, intensity, type, time, volume), measurement methods, methodological quality of studies, and type of Elisa kits used. Also, the absence of a significant difference in regular exercise intervention compared to acute exercise may be related to difficulties in controlling too many variables that may affect the results depending on the duration of the program. In this sense, Fleitas et al. [127] emphasized that some factors should be considered in the lack of significant effects of chronic physical exercise on peripheral BDNF levels. Failure to control participants’ nutritional status or energy intake during the study and before blood samples are collected may have a negative impact on BDNF results. Considering the included studies, it was observed that the energy intake of the participants during the exercise program was not considered in many studies. Conducting more controlled studies in the laboratory environment by applying standardized nutrition programs to the participants during the study may contribute to obtaining more accurate results. Additionally, circulating BDNF levels fluctuate throughout the day due to the diurnal variations of the cortisol and various hormonal fluctuations [128]. Thus, time of the day on both exercise interventions and data collection (blood collection time), and non-standardization of the collection may have a negative impact on BDNF-related measurements [127]. In this context, it is important to conduct quality and randomized controlled studies with larger sample groups, considering the energy intake of the participants in order to elevate circulating BDNF in response to exercise intervention in patients with obesity.

It is also important that authors of experimental studies accurately describe the research protocols, providing comprehensive information regarding the biological material and analytical procedures [129]. The different levels of BDNF are reported in serum and plasma. This can relate to thrombocytosis mechanism and utilization of BDNF after chronic, intensive, or long-term exercise. Higher production of growth factors in platelets and utilization of BDNF for reparation and regeneration of skeletal muscle and nerve fibers at the muscle levels is quite possible mechanism of BDNF reduction in serum and plasma [16,130].

Moreover, in our study, it was revealed that the circulating BDNF level did not differ significantly following the six months of high-intensity [70], and ten weeks of moderate-intensity [73] regular resistance exercises, and 6 months of moderate-intensity aerobic exercise [78]. In literature, studies have shown that resistance exercise had no significant effect on circulating BDNF concentration in both healthy [36,79,124], and elderly [127] individuals. Nevertheless, despite all these, it was seen that the circulating BDNF level increased after regular exercise in many studies included in present study. Regarding the studies in which the circulating BDNF level increased (1 study: moderate aerobic exercise, the other 7 studies: high-intensity aerobic exercise, a total of 8 studies), it was observed that high-intensity aerobic exercise performed for approximately 8–12 weeks were more likely to trigger the increase in circulating BDNF levels. In the light of this information, recent meta-analysis study suggested that moderate and high-intensity aerobic exercise programs (3–5 times a week and 20–60 min, 12 weeks) were an effective strategy to increase circulating BDNF levels in adolescents [131]. Therefore, aerobic exercise is effective in increasing BNDF level compared to resistance exercise. Indeed, regular exercise augmented circulating BDNF, while resistance exercise did not [79,124,127]. Aerobic exercise is linked to physiological processes that enhance BDNF, including improved endothelial function, insulin sensitivity, and cerebral blood flow [79,85,132]. Walsh et al. [85] suggested that high-intensity short-duration physical activities might increase circulating BDNF levels and improve brain health, and that combining aerobic and anaerobic exercise at approximately 60% of VO_2_max may have the greatest effect on circulating BDNF elevation due to mechanisms such as cardiovascular changes and lactate release.

The present meta-analysis study has limitations. Firstly, the potential confounding effect of age among participants included in the studies analyzed. Unfortunately, due to the small number of studies analyzing exercise-induced changes in circulating BDNF concentrations in obese subjects, subjects of different ages were included in the analysis. Moreover, age-related differences in physiological responses to exercise could impact the magnitude of BDNF upregulation, thereby influencing the overall effect size observed in our meta-analysis. Therefore, future studies with age-stratified analyses could provide more nuanced insights into the relationship between exercise, circulating BDNF, and obesity. Another limitation may be the gender of the subjects. Changes were not differentiated separately for male and female subjects. Older patients may exhibit differential responses to exercise interventions compared to younger individuals, which could influence the observed effects on circulating BDNF levels. Future investigations should consider stratified analyses based on gender, allowing for a more nuanced understanding of how exercise-induced changes in circulating BDNF levels may differ between male and female participants, especially in the context of obesity. Additionally, previous studies on different populations showed that duration of the exercise intervention (weeks) [83,125,126,127], intensity of exercise [79,85], the weekly volume of the exercise [126] did not significantly affect the circulating BDNF level. Although the studies above do not show a significant effect, the last limitation in our study is that, due to the small number of studies in obese individuals, separate submodular analyzes were not performed to examine the effects of individual components of exercise prescription such as time, intensity, type, and frequency on circulating BDNF responses. These components are critical determinants of the physiological responses to exercise, and their nuanced exploration could provide valuable insights into the specific parameters influencing circulating BDNF levels in individuals with obesity. For instance, different exercise modalities, such as aerobic, resistance, or combined training may elicit distinct molecular responses, and the variability in exercise interventions could introduce heterogeneity into our meta-analysis. Lastly, BDNF’s pivotal role in brain development, cognition, and mood regulation is extensively documented, particularly in adults with obesity. Studies indicate that circulating BDNF responses to stimuli such as exercise vary across different life stages. This emphasis may arise from the marked physiological and developmental differences between adults and children. While childhood obesity remains a significant health concern, included studies in our meta-analysis concentrates on exploring circulating BDNF responses to exercise in especially adults with obesity. Although there are relatively few studies in children in the literature, children’s circulating BDNF levels and responses to exercise may diverge significantly from those of adults due to ongoing growth and maturation processes. Nevertheless, we recognize the significance of addressing childhood obesity and its enduring impact on neurodevelopment. Future investigations could specifically target childhood obesity and examine in depth how exercise modulates circulating BDNF levels in this population only, tailoring interventions to accommodate developmental stages and foster healthy brain development from an early age. Such a nuanced approach promises to enhance our understanding of BDNF dynamics across the lifespan and guide targeted strategies for managing obesity and promoting brain health. Recognizing this limitation is essential for interpreting our findings and emphasizing the need for more focused investigations into the specific effects of different exercise programs on circulating BDNF in the context of obesity. Thus, more studies are needed to provide a clearer understanding of which type of physical activity has the greatest impact on circulating BDNF in patients with obesity. By acknowledging these limitations, we aim to provide a comprehensive and transparent assessment of the findings while highlighting areas for future research to address these potential sources of bias and variability. Well designed and quality randomized controlled studies with greater populations are recommended to determine how different exercise configurations may affect circulating BDNF levels in patients with obesity [126].

## 5. Conclusions

The present meta-analysis represents a pioneering effort in investigating the impact of acute and regular physical activity on circulating BDNF levels, focusing exclusively on individuals with obesity. This study distinguishes itself by synthesizing findings exclusively from randomized control group studies, thereby contributing a specialized perspective to the existing literature. The principal objective was to discern whether acute exercise or a more regular physical exercise could elicit alterations in circulating BDNF levels in comparison to a control group within the context of obesity. The meta-analysis reveals that acute exercise induces a noteworthy elevation in circulating BDNF levels among individuals with obesity when compared with the control group. However, intriguingly, regular physical exercise did not yield a statistically significant alteration in circulating BDNF levels when compared to the control group. Although the observed effect of regular physical exercise interventions on circulating BDNF did not attain statistical significance, there was a discernible small-scale impact, underscoring the nuanced nature of the relationship. This investigation underscores the potential utility of both acute and regular exercise regimens in positively modulating circulating BDNF levels in the context of obesity. Given the intricate interplay of circulating BDNF with appetite regulation, neuroplasticity, and cognitive functions, augmenting the diminished basal levels inherent in obesity through non-pharmacological approach, such as physical exercise, emerges as a promising avenue for mitigating the condition. Moreover, the potential ancillary benefits extend beyond mere weight management, encompassing the amelioration of psychological comorbidities associated with obesity, including cognitive disorders and depression. Additionally, given the complex processes regulating BDNF levels in tissues, it is challenging to explain the relationship between BDNF and obesity. Therefore, further research is needed to understand the mechanisms of exercise and BDNF regulation to develop more effective therapies for obesity and optimize exercise-based treatments. Consequently, the study underscores the importance of customizing exercise recommendations for managing obesity and emphasizes the necessity of tailored interventions to maximize neurotrophic responses. Future research should concentrate on understanding how the body adapts to exercise and exploring innovative approaches to enhance circulating BDNF regulation through physical activity in individuals with obesity.

Considering the mechanisms of action of BDNF and the multitude of factors influencing the secretion of this protein in healthy people, in obesity, and under the influence of physical exercise, it is important for the authors of future studies to prepare detailed descriptions of experimental studies. Our meta-analysis indicates the need for precise characteristics of the studied patient groups and the conditions of conducting exercise tests, including a detailed description of the laboratory and environmental test conditions. This will enable certain conclusions to be drawn regarding the results of meta-analyses in the future.

## Figures and Tables

**Figure 1 biology-13-00323-f001:**
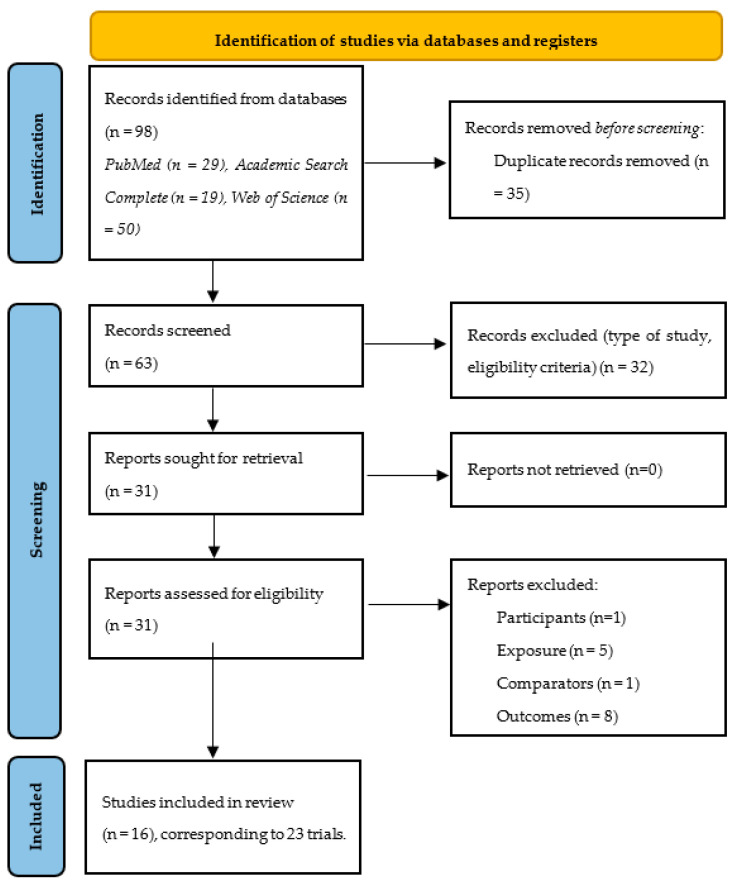
PRISMA 2020 flow diagram.

**Figure 2 biology-13-00323-f002:**
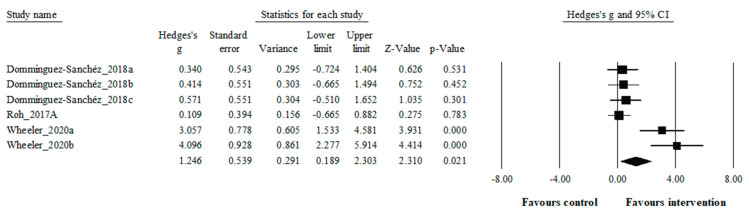
Forest plot illustrating the effect of acute exercise on circulating BDNF levels in comparison to controls. Forest plot values are shown as effect sizes (Hedges’ g) with 95% confidence intervals (CI). Black squares: individual studies. The size represents the relative weight. Black rhomboid: summary value. Mean results: ES = 1.25, 95% CI = 0.19 to 2.30, *p* = 0.021, I^2^ = 80.4%, N total participants = 104 [63,64,65].

**Figure 3 biology-13-00323-f003:**
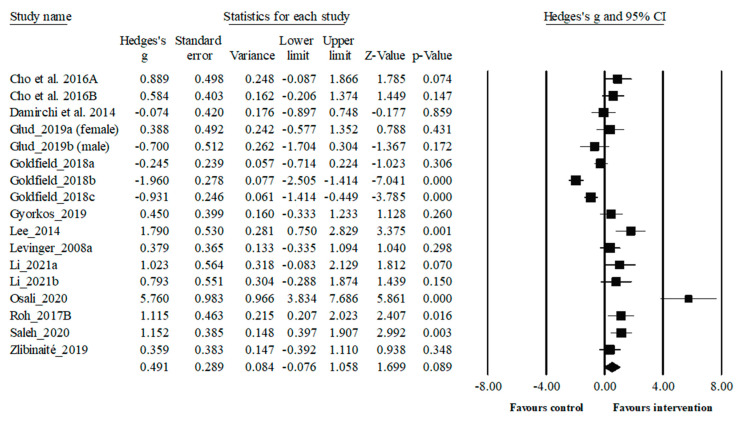
Forest plot illustrating the effect of regular exercise intervention on circulating BDNF levels in comparison to controls. Forest plot values are shown as effect sizes (Hedges’ g) with 95% confidence intervals (CI). Black squares: individual studies. The size represents the relative weight. Black rhomboid: summary value. Mean results: ES = 0.49, 95% CI = −0.08 to 1.06, *p* = 0.089, I^2^ = 88.7%, N total participants = 571 [66,67,68,69,70,71,72,73,74,75,76,77,78].

**Table 1 biology-13-00323-t001:** Full search strategy for each database.

Database	Specificities of the Databases	Search Strategy
PubMed	None to report	(“brain derived neurotrophic factor”[MeSH Terms] OR (“brain derived”[All Fields] AND “neurotrophic”[All Fields] AND “factor”[All Fields]) OR “brain derived neurotrophic factor”[All Fields] OR “bdnf”[All Fields] OR (“brain derived neurotrophic factor”[MeSH Terms] OR (“brain derived”[All Fields] AND “neurotrophic”[All Fields] AND “factor”[All Fields]) OR “brain derived neurotrophic factor”[All Fields] OR (“brain”[All Fields] AND “derived”[All Fields] AND “neurotrophic”[All Fields] AND “factor”[All Fields]) OR “brain derived neurotrophic factor”[All Fields])) AND (“aerobic*”[All Fields] OR “HIIT”[All Fields] OR (“high intensity interval training”[MeSH Terms] OR (“high intensity”[All Fields] AND “interval”[All Fields] AND “training”[All Fields]) OR “high intensity interval training”[All Fields] OR (“high”[All Fields] AND “intensity”[All Fields] AND “interval”[All Fields] AND “training”[All Fields]) OR “high intensity interval training”[All Fields]) OR “anaerobic*”[All Fields]) AND (“obeses”[All Fields] OR “obesity”[MeSH Terms] OR “obesity”[All Fields] OR “obese”[All Fields] OR “obesities”[All Fields] OR “obesity s”[All Fields] OR (“overweight”[MeSH Terms] OR “overweight”[All Fields] OR “overweighted”[All Fields] OR “overweightness”[All Fields] OR “overweights”[All Fields]))
Academic Search Complete	Search for title and abstract also includes keywords	“(BDNF OR brain-derived neurotrophic factor) AND (aerobic* OR HIIT OR high intensity interval training OR anaerobic*) AND (obesity OR overweight)
Web of Science	Search for title and abstract also includes keywords and its designated “topic”	BDNF OR brain-derived neurotrophic factor (All Fields) and aerobic* OR HIIT OR high intensity interval training OR anaerobic* (All Fields) and obesity OR overweight (All Fields)

**Table 2 biology-13-00323-t002:** Characteristics of the included studies for acute effect on circulating BDNF level (N: 3, 6 trials).

Study	Study Design	n	Sex	Groups	Acute Exercise Protocol	Main Outcome	Brand/Company Name of BDNF Kits	Main Results (BDNF)
Dominguez-Sanchez 2018 [63]	RCT	51	Physically inactive Men	-HIIT: 14 -RT: 12 -CT: 12 -CG: 12	HIIT: 4 × 4-min intervals at 85–95% HRmax with 4-min active recovery at 75–85% HRmax RT: ≈12–15 reps per set of six exercises targeting major muscle groups at high intensity. CT: underwent both the HIIT and RT protocols CG: Non-exercising	Plasma BDNF (ng/mL)	SPR Biosensors methods, an amino-coupling chemistry kit KAN-50 Coupling Kit (GE Healthcare, Uppsala, Sweden)	-HIIT: Increased (+6.8%, *p* = 0.134) -RT: Increased (+9.3%, *p* = 0.066) -CT: Increased (+11.6%, *p* < 0.05) -CG: Increased (+0.6%, *p* = 0.804)
Roh et al., 2017 [64]	RCT	24	Untrained Men	-Ob: 12 -NonOb: 12	Treadmill run of 20 min, %85 VO_2_max	Serum BDNF (pg/mL)	Human BDNF ELISA kit (cat. no. DBD00; R&D Systems, Minneapolis, MN, USA)	- Increased (Ob > NonOb)
Wheeler et al., 2020 [65]	RCT	65	Sedentary men and women	-SIT: 22 - EX + SIT: 23 - EX + BR: 20	SIT: 8 h uninterrupted sitting EX + SIT: 1 h sitting, 30 min walking (65–75% HRmax), 6.5 h uninterrupted sitting. EX + BR: 1 h sitting, 30 min walking (65–75% HRmax), 6.5 h sitting interrupted every 30 min with 3 min light-intensity walking.	Serum BDNF (ng/mL)	Human BDNF ELISA Kits (R&D Systems, Wiesbaden, Germany	- EX + SIT: increased (+171), *p* < 0.05 - EX + BR: increased (+139), *p* < 0.05 -SIT: Decreased (−227)

Abbreviations: n = Population; BDNF = Brain-derived neurotrophic factor; RCT = Randomized controlled trial; BMI = Body mass index; HIIT = High intensity interval training; RT = Resistance training; CT = Combined training; EX + SIT = exercise + sitting; EX + BR = exercise + breaks; SIT = Sitting; BR: Breaks; CG = Control group; Ob = Obese; NonOb = Non-obese; HRmax = Maximum heart rate; VO_2_max = Maximum oxygen intake; Hr: Hours; SPR: Surface Plasmon Resonance, Elisa: Enzyme-linked immunosorbent assay.

**Table 3 biology-13-00323-t003:** Characteristics of the included studies for regular exercise on circulating BDNF level (N: 13, 17 trials).

Study	Study Design	*n*	Sex	Groups	Regular Exercise Protocol	Main Outcome	Brand/Company Name of BDNF Kits	Main Results (BDNF)
Cho et al., 2016 [66]	RCT	16	Physically inactive men	-EG: 8 -CG: 8	EG: Supervised treadmill running at 70% of HRR, 40 min each session, 3 times a week for 8 weeks. CG: maintained their own life-styles with no intervention	Serum BDNF (ng/mL)	A human BDNF ELISA Kit (R&D Systems, Minneapolis, MN, USA).	-EG: Increased (20.56%), *p* < 0.05
Cho et al., 2016 [67]	RCT	36	Physically inactive women	-AE: 12 -AE + CES: 12 -CG: 12	AE: Three times 40-min treadmill running sessions per week for 8 weeks at 70% of HRR. CG: maintained their own life-styles with no intervention	Serum BDNF (pg/mL)	A human BDNF ELISA Kit (Cat. no. DBD00; R&D Systems, Minneapolis, MN, USA)	-AE: Increased (AE > CG), *p* < 0.05
Damirchi et al., 2014 [68]	RCT	21	Physically inactive men with MetS	-EG: 11 -CG: 10	EG: a 6-week aerobic training: (3 sessions per week; 25–40 min walking, running by 50–60% of $\overset{\cdot }{V}$O_2_peak) CG: maintained their own life-styles with no intervention	Serum BDNF (pg/mL)	A human BDNF ELISA Kit (R&D Systems, Minneapolis, MN, USA)	-EG: Decreased, *p* < 0.05
Glud et al., 2019 [69]	RCT	50	Physically inactive men and women	-EXO: 7 women, 9 men -DIO: 8 women, 6 men -DEX: 11 women, 9 men	EXO: 12 weeks of aerobic exercise and isocaloric diet. Supervised aerobic exercise 3 times/week, 60–75 min/session, 500–600 kcal/session, intensity at 70% of HRR. DEX: 12 weeks of aerobic exercise alongside 8 weeks of VLED (800 kcal/day), followed by a 4-week weight maintenance diet.	Serum BDNF (ng/mL)	Quantikine ELISA Human Free BDNF immunoassay (DBD00, R&D Systems, Abingdon OX14, UK)	-EXO: decreased (22.4%, *p* < 0.05) in women, (22.1%, *p* < 0.05) in men -DIO: decreased (29.9%, *p* < 0.05) in women, and (4.2%, *p* < 0.05) in men -DEX: decreased (32.5%, *p* < 0.05) in women.
Goldfield et al., 2018 [70]	RCT	282	Irregularly active men and women	-AE: n = 69 -RT: n = 70 -CT: n = 74 -CG: n = 69	AE: Aerobic exercise on treadmills, elliptical machines, and/or bicycle ergometers, 6-month intervention, twice a week, 20–45 min/session, 65–85% HRmax. RT: Resistance training, 6-month intervention, twice a week, progressing from 20 to 45 min/session. Exercises using weight machines or free weights, progressing from 2 sets of 15 reps at moderate intensity to 3 sets of 8 reps at 8-RM. CT: Combination of AE and RT. CG: maintained their own life-styles with no intervention	Serum BDNF (ng/mL)	Human Free BDNF Quantikine ELISA kit, R&D systems, Cat# DBD00)	-AE group: Increased (+1.80) -RT group: Decreased (−2.00) -CT group: Decreased (−1.70)
Gyorkos et al., 2019 [71]	RCT	12	Sedentary free-living individuals	-CRPD-Sed: 5 -CRPD-Ex: 7	CRPD-Ex: HIIT on a cycle ergometer. Three min warm-up, 10 × 60 s cycling intervals with 60 s active recovery, ~90% HRmax, and a 3 min cool down. 3 sessions per week for four weeks. CRPD-Sed: (<50 g Carbohydrate)	Serum BDNF (ng/mL)	Human BDNF Elisa Kit (#DBD00, Thermo Fisher Scientific)	-CRPD-Sed: Increased (+20%), *p* < 0.05 -CRPD-Ex: Increased (+38%), *p* < 0.05 -CRPD-Ex > CRPD-Sed.

Abbreviations: n = Population; BDNF = Brain-derived neurotrophic factor; RCT = Randomized controlled trial; MetS = Metabolic syndrome; EG = Exercise group; RT = Resistance training; CT = Combined training; HIIT = High Intensity Interval Training; RM = Repetition maximum; CG = Control group; EXO = Exercise-only; DIO = Diet-only; DEX = Diet and exercise; AE = Aerobic exercise; CES = Cranial electrotherapy stimulation; CRPD-Sed = Carbohydrate-restricted paleolithic-based diet with sedentary behavior; CRPD-Ex = Carbohydrate-restricted Paleolithic-based diet with high intensity interval training; HRR = Heart rate reserve; $\overset{\cdot }{V}$O_2_peak = Highest oxygen uptake; VLED = very low energy diet; Elisa: Enzyme-linked immunosorbent assay.

**Table 4 biology-13-00323-t004:** Characteristics of the included studies for regular exercise on circulating BDNF level (N: 13, 17 trials) (Continuation of Table 3).

Study	Study Design	n	Sex	Groups	Regular Exercise Protocol	Main Outcome	Brand/Company Name of BDNF Kits	Main Results (BDNF)
Lee et al., 2014 [72]	RCT	26	Physically inactive men and women	-Ob: 8 -T2DM: 7 -CG: 11	AE: 40–60 min per session at 50~60% VO_2_max, 3 sessions a week, for 12 weeks. CG: maintained their own life-styles with no intervention	Serum BDNF (ng/mL)	A human BDNF Elisa Kit (R&D Systems, Minneapolis, MN, USA)	Ob: Increased
Levinger et al., 2008 [73]	RCT	49	Physically inactive men and women	-HiMF-Exp: 15 -HiMF-CG: 14 -LoMF-Exp: 10 -LoMF-CG: 10	RT: 10 weeks. Initially, two sets of 15–20 reps at 40–50% 1RM. From weeks 2–10, progressed to three sets, 8–20 reps, at 50–85% 1RM. CG: maintained their own life-styles with no intervention	Serum BDNF (pg/mL)	Human BDNF Elisa Kit (Catalog number: DY248; Minneapolis, MN, USA)	-RT: Unchanged
Li et al., 2021 [74]	RCT	29	Physically inactive men and women	-HIIT: 10 -VICT: 10 -CG: 9	HIIT: 4 × 3 min at 90% VO_2_max with 3 min at 60% VO_2_max, about 45 min/session, 3 sessions per week for 12 weeks. VICT: 25 min at 70% VO_2_max, about 45 min/session, 3 sessions per week for 12 weeks. CG: maintained their own life-styles with no intervention	Serum BDNF (pg/mL)	Human BDNF Elisa Kit (Abcam Inc., Cambridge, UK)	- HIIT: Increased, *p* < 0.05 - VICT: Increased, *p* < 0.05 - There was no significant difference between HIIT and VICT in terms of BDNF.
Osali et al., 2020 [75]	RCT	44	Physically inactive women	-MetS exercise + Nano-Curcumin: 11 -MetS exercise: 11 -MetS Nano-Curcumin: 11 -MetS CG: 11	AE: moderate intensity (65–75% HRR) on a treadmill (run or walk) for 3 sessions per week, each lasting 12–17 min, over 6 weeks. CG: maintained their own life-styles with no intervention	Serum BDNF (pg/mL)	Human BDNF Elisa Kits (Adipo Bioscience, Santa Clara, CA, USA)	EG: Increased, *p* < 0.05
Roh et al., 2017b [76]	RCT	20	Physically inactive men	-Ob: 10 -NonOb: 10	AE: 40 min, 3 times a week, for 8 weeks, at 70% HRR, totaling 60 min per exercise session	Serum BDNF (pg/mL)	Human BDNF ELISA Kit (#DBD00; R&D Systems, Minneapolis, MN, USA)	Ob: Increased, *p* < 0.05 NonOb: Unchanged
Saleh et al., 2020 [77]	RCT	60	Physically inactive men and women	-Ob-Exp: 15 -Ob-CG: 15 -NW-Exp: 15 -NW-CG: 15	Anaerobic gymnastics exercise: 45 min/session, 3 sessions per week for 8 weeks. CG: maintained their own life-styles with no intervention	Serum BDNF (pg/mL)	Human BDNF PicoKine™ ELISA Kit (Catalog No. EK0307; R&D Systems, Austria)	Ob: Increased (+33.80%), *p* < 0.05 NW:Increased (+31.36%, *p* < 0.05
Zibinaite et al., 2019 [78]	RCT	26	Sedentary women	EG: 13 CG: 13	AE: 72 supervised exercise sessions on cycle ergometers over 6 months, 3 sessions per week. Each session lasted 50 min at an intensity between 60% and 70% of HRmax. CG: maintained their own life-styles with no intervention	Serum BDNF (pg/mL)	Human BDNF Elisa Kits (Gemini; Stratec Biomedical, Birkenfeld, Germany).	EG: Unchanged CG: Unchanged

Abbreviations: n = Population; BDNF = Brain-derived neurotrophic factor; RCT = Randomized controlled trial; MetS = Metabolic syndrome; T2DM = Type 2 diabetes mellitus; HiMF = number of metabolic risk factors ≥ 2; LoMF = number of metabolic risk factors ≤ 1; HIIT = High intensity interval training; VICT = Vigorous-intensity continuous training; AE = Aerobic Exercise; 1RM = One-repetition maximum; VO_2_max = Maximum oxygen intake; HRmax = Maximum heart rate; HRR = Heart rate reserve; RT = Resistance Training; EG = Exercise group; CG = Control Group; Ob = Obese; NonOb = Non-obese; NW = Normal Weight; Exp = Experimental; Elisa: Enzyme-linked immunosorbent assay.

## Data Availability

Data are available for research purposes upon reasonable request to the corresponding author.

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
