# Peer review of "Exploring the Effect of Acute and Regular Physical Exercise on Circulating Brain-Derived Neurotrophic Factor Levels in Individuals with Obesity: A Comprehensive Systematic Review and Meta-Analysis"

_biology, 2024, doi:10.3390/biology13050323_

Round 1

Reviewer 1 Report

Comments and Suggestions for Authors

Title: levels in where? Revise the title.

Abstract: Authors should start with stating where BNDF quantification performed. Authors should rewrite 18-25 it is hard to follow.

Introduction: Authors should add background and rationale about why they quantified BDNF? What is the importance of that quantification.

Authors should add rationale about how their experiments will provide valuable information and what will be main aims possibly related with their experiment.

Results: Authors should avoid making deductions in the results section. They only need to report and strictly articulate their results

Authors should revise tables, they are not easy to follow

Authors should provide abbreviation section separately for ease

Synthesis of results not proper word

Discussion: Authors should give information about what is currently know in the field and how their results fitting with them instead of ordering experimental results from different studies.

Discussion seems okay, authors should state how their experimental results will help for future studies, how it will help to understand for basic mechanisms related with obesity and physical exercise

Author Response

Reviewer 1

Title: levels in where? Revise the title.

Response: Dear Reviewer, thank you for your suggestion. It was revised.

Abstract: Authors should start with stating where BNDF quantification performed. Authors should rewrite 18-25 it is hard to follow.

Response: Thank you for noticing that. The abstract was re-written for a better clarity. We have taken your request into consideration.

Introduction: Authors should add background and rationale about why they quantified BDNF? What is the importance of that quantification.

Response: Dear Reviewer, thank you for noticing that. The authors totally agree with you. We expanded introduction session to increase the background for this review. 

Authors should add rationale about how their experiments will provide valuable information and what will be main aims possibly related with their experiment.

Response: Considering the comment above about this session, we tried to highlight the importance of having good values of BDNF for neurological health, especially for memory and learning. We improved the introduction. Thank you so much, Dear Reviewer.

Results: Authors should avoid making deductions in the results section. They only need to report and strictly articulate their results

Response: Thank you so much, Dear Reviewer. Statements regarding deductions have been deleted. We revised the results section.

Authors should revise tables, they are not easy to follow

Response: Thank you so much, Dear Reviewer. We reduced the number of tables to make them more understandable. All tables have been revised and re-created.

Authors should provide abbreviation section separately for ease

Response: Thank you so much, Dear Reviewer. To make it more understandable, we created the Abbreviations section.

Synthesis of results not proper word

Response: Thank you for your observation. Section 3.3. was amended accordingly. In the revised version we wrote “Results of syntheses”. Moreover, subheadings in the method and results section were prepared taking into account the title in Prisma Checklist 2020. Thank you.

Discussion: Authors should give information about what is currently know in the field and how their results fitting with them instead of ordering experimental results from different studies.

Response: Dear Reviewer. Thank you. We have made correction in the discussion part regarding your suggestions.

Discussion seems okay, authors should state how their experimental results will help for future studies, how it will help to understand for basic mechanisms related with obesity and physical exercise

Response: Dear Reviewer. Thank you. We have added suggested text.

Reviewer 2 Report

Comments and Suggestions for Authors

The research titled "The study entitled "Exploring the Effect of Acute and Regular Physical Exercise on Brain-Derived Neurotrophic Factor Levels in Individuals with Obesity: A Comprehensive Systematic Review and Meta-Analysis" seems promising and holds potential for enhanced therapeutic outcomes. However, there are a few questions that require clarification.

1. The author should include background information on Brain-Derived Neurotrophic Factor (BDNF) and its relationship with obesity in the Abstract section.

2. The Abstract section requires refinement for clarity and structure. The author should emphasize the study's novelty, address its limitations, and summarize key findings more effectively.

3. The significance of increased BDNF levels in obesity following exercise needs to be highlighted by the author in the Abstract section.

4. The Introduction section lacks discussion on previous literature regarding BDNF's role in mitigating neurological impairments associated with obesity.

5. The rationale behind focusing on participants aged 18 and above needs clarification, as well as how childhood obesity is addressed in the analysis.

6. Subheadings need to be revised for better clarity and detail.

7. The manuscript contains spelling errors and requires grammatical checking.

8. The authors need to explain how they justified including participants with comorbidities alongside obesity in their criteria for inclusion.

9. Explanation is needed regarding how the authors categorized obese participants and why they chose not to use any other scale.

10. More details on the types of acute and regular exercise studied can be provided by the authors.

11. The authors assert the circulating BDNF levels, yet the inclusion criteria lack specific information regarding the sample.

12. The inclusion and exclusion criteria could be more elaborately described to provide a clearer understanding of participant selection for the study.

13. Did the authors include clinical trial studies in their participant inclusion criteria?

Comments on the Quality of English Language

The language of the manuscript needs minor revision. 

Author Response

The research titled "The study entitled "Exploring the Effect of Acute and Regular Physical Exercise on Brain-Derived Neurotrophic Factor Levels in Individuals with Obesity: A Comprehensive Systematic Review and Meta-Analysis" seems promising and holds potential for enhanced therapeutic outcomes. However, there are a few questions that require clarification.

Response: Dear Reviewer, thank you for your kind words. The authors made an effort to improve the manuscript considering your comments and suggestions.

  1. The author should include background information on Brain-Derived Neurotrophic Factor (BDNF) and its relationship with obesity in the Abstract section.

Response: Dear Reviewer, thank you for your comment. We added this information in the abstract. We improved the introduction section.

  1. The Abstract section requires refinement for clarity and structure. The author should emphasize the study's novelty, address its limitations, and summarize key findings more effectively.

Response: Dear Reviewer, thank you for your suggestion. The authors tried to improve the clarity of this session. We have revised the abstract section taking your requests into consideration.

  1. The significance of increased BDNF levels in obesity following exercise needs to be highlighted by the author in the Abstract section.

Response: Thank you for your comment. We changed accordingly.

  1. The Introduction section lacks discussion on previous literature regarding BDNF's role in mitigating neurological impairments associated with obesity.

Response: Dear Reviewer. Thank you for your comment. We reformulated this session, including some new information and studies regarding BDNF's role in mitigating neurological impairments associated with obesity at the end of the introduction.

  1. The rationale behind focusing on participants aged 18 and above needs clarification, as well as how childhood obesity is addressed in the analysis.

Response: For a better clarity, it was added in the methods section that children and adolescents were excluded since “BDNF levels change dramatically over development and in response to changes in circulating gonadal hormones (Bath et al., 2013)”. Also, we added one paragraph at the end of the discussion section (limitations section). Necessary explanations and suggestions were made in the limitations section. Thank you so much for your contributions, Dear Reviewer.

  1. Subheadings need to be revised for better clarity and detail.

Response: Dear Reviewer, Thank you so much. We changed the subheadings. Moreover, Subheadings in the method and results section were prepared taking into account the titles in Prisma Checklist 2020. Thank you.

  1. The manuscript contains spelling errors and requires grammatical checking.

Response: The manuscript was revised accordingly. Thank you.

  1. The authors need to explain how they justified including participants with comorbidities alongside obesity in their criteria for inclusion.

Response: Dear Reviewer, thank you for your comment. There are limited studies in the literature examining the effect of exercise on BDNF in obese individuals. Unfortunately, it is common for obese individuals to develop more comorbidities. Therefore, they were included to achieve a more accurate representation of the target population.

  1. Explanation is needed regarding how the authors categorized obese participants and why they chose not to use any other scale.

Response: Dear Reviewer, thank you for your comment. The chosen categorization was based on the inclusion of this analysis in the majority of studies. We acknowledge that BMI may not always be the optimal tool for assessing body composition, as not all studies incorporated more reliable measurements like DEXA or skinfold assessments. This information was included in the manuscript.

  1. More details on the types of acute and regular exercise studied can be provided by the authors.

Response: Dear Reviewer, thank you for your comment. More information was added accordingly.

  1. The authors assert the circulating BDNF levels, yet the inclusion criteria lack specific information regarding the sample.

Response: Dear Reviewer, thank you for your comment. Serum and plasma BDNF values were taken into account. It was added to the inclusion criteria in the methods section.

  1. The inclusion and exclusion criteria could be more elaborately described to provide a clearer understanding of participant selection for the study.

Response: Dear Reviewer, the authors tried to complete and clarify the inclusion and exclusion criteria for a better understanding.

  1. Did the authors include clinical trial studies in their participant inclusion criteria?

Response: Dear Reviewer, thank you for your comment. More information was added accordingly.